# BEYOND DATA SILOS : LEVERAGING DISPARATE DATA SOURCES FOR ITE ESTIMATION

## ABSTRACT

While ML systems for Individual Treatment Effect (ITE) estimation have advanced healthcare decision-making, conventional methods require substantial amounts of costly training data for each intervention under consideration. In this work, we present a novel framework, based on causal transformers, for collaborative learning of heterogeneous ITE estimators across disparate data sources. Our approach can be deployed across distributed institutions (such as hospitals) via Federated Learning, enabling training on a large and diverse dataset (without sharing sensitive health data), and the same framework can be applied locally when multiple heterogeneous data sources exist within a single institution, breaking down data silos. The proposed method is flexible to handle diverse patient populations and non-identical patient measurements (covariates) across different data sources, while allowing for the estimation of treatment effects of disparate treatments being administered across these sources. Moreover, this framework can be utilized to predict the effects of novel and unseen treatments by utilizing available treatment level information. Thorough experimental evaluation on real-world clinical trial and widely-used research datasets demonstrates that our method surpasses existing baselines. Furthermore, analysis of our model's attention mechanisms reveals clinically meaningful disease and treatment-related patterns validated by domain expertise, demonstrating the interpretability and clinical relevance of our approach.

## 1 INTRODUCTION

Understanding the impact of an intervention on the outcome, also known as the treatment effect, is essential for identifying causation and treatment selection in clinical decision-making. Since interventions often produce different effects on different individuals, determining the extent to which diverse individuals respond to interventions, known as *Individual Treatment Effects (ITE)* estimation, is an important problem. Methodologically, ITE estimation relies on counterfactual analysis, which predicts the outcomes for individuals who have undergone a treatment (factual) under different exposure scenarios (counterfactual) that they haven't actually encountered (Neyman, 1923; Rubin, 2005).

In recent years, several data-driven machine learning approaches have been extensively utilized for this purpose (Johansson et al., 2016; Shalit et al., 2017; Curth & van der Schaar, 2021a;b; Bica & van der Schaar, 2022a; Curth & Van Der Schaar, 2023; Guo et al., 2023; Xue et al., 2023). However, these methods face limitations in practice as estimating individual treatment effects from a single data source is constrained by insufficient sample sizes per intervention, especially for rare conditions or treatments. This necessitates leveraging multiple data sources, but since clinical trials and observational studies are designed and conducted independently, each with distinct research objectives, inclusion criteria, and measurement protocols, there are significant challenges.

First, when data exists across institutions, privacy regulations and data sharing restrictions prevent centralization. Federated Learning (FL) (McMahan et al., 2017) offers a potential solution by enabling organizations to collaborate while maintaining local data privacy. Second, and more fundamental, is the substantial heterogeneity inherent in multi-source data, whether across institutions or within a single institution's disparate systems. This heterogeneity manifests through differences in patient populations, measurement protocols, variable definitions, and treatment practices, creating inconsistent feature and treatment spaces. For example, Hospital 1 might administer Treatment A and collect variables {age, blood pressure, biomarker P} while Hospital 2 uses Treatment B and records {age, weight, biomarker R}. This heterogeneity creates partially overlapping but fundamentally

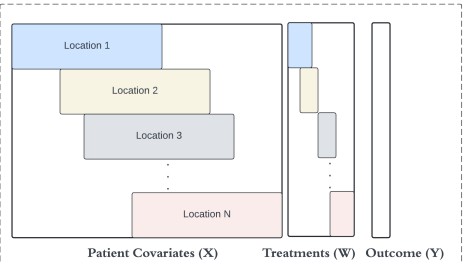

Figure 1: An overview of the setting with N locations, where each location has some overlap with the other locations in their non-identical feature as well as the treatment space.

incompatible data sources, as illustrated in Figure 1, posing a challenge for conventional FL. Previous approaches to federated ITE estimation (Vo et al., 2022; Xiong et al., 2023) inadequately address this complexity, as they predominantly assume identical treatments and feature spaces across data sources, assumptions that rarely hold in real-world settings. Notably, similar heterogeneity challenges can also arise within single institutions where legacy systems, departmental divisions, or acquisition histories have created internal data silos with inconsistent variable definitions and treatment records.

In this work, we propose *Federated Transformers for Treatment Effects Estimation (FedTransTEE)*, an end-to-end framework for ITE estimation across institutions that addresses the challenges of heterogeneous feature, treatment, and outcome spaces. FedTransTEE constructs personalized solutions for each institution while leveraging shared data patterns across sites. The framework employs a transformer-based covariate encoder to learn a common representation space for covariates, accommodating non-identical feature sets across institutions. Intervention embeddings, capturing intervention-specific features, are learned separately and combined with patient embeddings via a cross-attention transformer. A personalized predictor then estimates the ITE for interventions unique to each site. Moreover, by incorporating intervention-specific features or descriptions, the framework enables generalization to unseen interventions in zero-shot settings, making it capable of reliably forecasting the effects of newly designed therapies.

To the best of our knowledge, this is the first work that specifically addresses federated treatment effects estimation across multiple institutions under heterogeneous covariate, treatment, and outcome spaces. Our **key contributions** are:

1. We propose *FedTransTEE*, a novel causal transformer-based framework for ITE estimation that systematically accommodates heterogeneous covariates, treatments, and outcome spaces, to enable learning across disparate data sources, whether unifying internal data silos within a single institution or connecting distributed institutions via federated learning, making a significant advancement in treatment effect estimation for healthcare.

2. The framework also introduces a zero-shot ITE estimation capability, allowing for the prediction of treatment effects for previously unseen therapies. This advancement opens new possibilities for novel therapy evaluation and clinical trial planning.

3. Explainable and actionable insights are provided through attention mechanisms, enabling interpretable treatment decisions and enhancing trust in clinical applications. These insights are validated by a domain expert (specialized in stroke), emphasizing their clinical relevance.

4. We validate the framework on several research and real-world clinical trial datasets, demonstrating its robustness to heterogeneity and superior performance compared to state-of-the-art baselines. These evaluations highlight the practical utility of FedTransTEE in addressing complex, real-world challenges in healthcare.

## 2 BACKGROUND

The ITE estimation problem refers to prediction of the effects of different interventions on individual subjects. A specific intervention data is denoted as $\boldsymbol{D} = (\mathcal{X}, \mathcal{Y}, \mathcal{T})$, where $\mathcal{X}$ encodes the pre-intervention covariate vectors of the subjects, $\mathcal{T}$ encodes the intervention (or treatment), and $\mathcal{Y}$ denotes the outcome corresponding to the intervention. Without loss of generality, we consider a $K$ intervention setting, wherein $\mathcal{T} \in \{0, 1, \ldots K\}$ with $\mathcal{T} = 0$ denoting no intervention and $\mathcal{T} = j$ denoting use of the $j^{th}$ intervention. We use notation $\mathcal{Y}(0)$ to record the outcome under placebo or

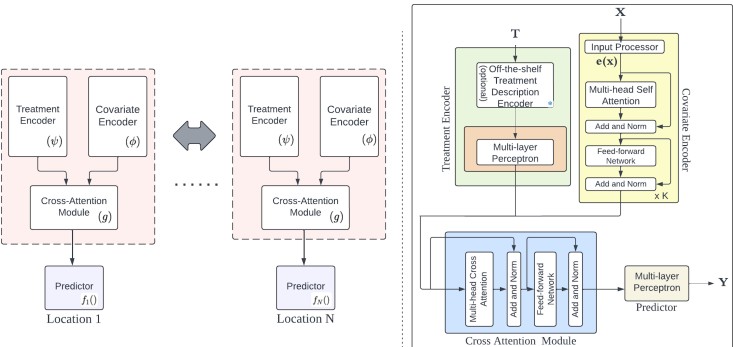

Figure 2: The above figure illustrates the FedTransTEE framework, with $N$ site locations on the left; as depicted, the covariate encoder, the treatment encoder and the cross-attention module are shared across locations but the predictor is personalised for each location. A detailed view of the model architecture with specifics of each component is shown on the right side.

standard-of-care intervention and $\mathcal{Y}(j)$ to record the outcome under the $j^{th}$ intervention. For each subject in the population, we only see one of the outcomes associated with the intervention that was used on the subject. The probability of the intervention assignment or the propensity score is denoted by $\pi(x, j) = \mathbb{P}(\mathcal{T} = j | \mathcal{X} = x)$. The Neyman-Rubin (Neyman, 1923; Rubin, 2005) framework for causal inference suggests designing a separate potential outcome function for each intervention that can be applied on individual subjects, $\mu_j(x) = \mathbb{E}(\mathcal{Y} | \mathcal{X} = x, \mathcal{T} = j)$. These functions are then used to predict the effect of the treatment $j$.

The framework operates under three common causal inference *assumptions - Stable Unit Treatment Value Assumption (SUTVA)*, *unconfoundedness*, and *stochasticity*. Under these assumptions, the ITE of an intervention $j$ is approximated by the Conditional Average Treatment Effect (CATE), $\tau_j(x) = \mu_j(x) - \mu_0(x)$. A learning model estimates the CATE by predicting $\hat{\mu}_j(x)$ and $\hat{\mu}_0(x)$, denoted by $\hat{\tau}_j(x) = \hat{\mu}_j(x) - \hat{\mu}_0(x)$. Additional details on assumptions are provided in Appendix B.

Now, consider an FL setting with $N$ clients (institutions or hospitals). Each client $m$ maintains a local dataset $\boldsymbol{D}_m = (\mathcal{X}^m, \mathcal{Y}^m, \mathcal{T}^m)$ with $n_m$ subjects, $d_m$-dimensional covariates, multiple interventions $\mathcal{T}^m$, and corresponding factual outcomes $\mathcal{Y}^m$. The heterogeneity manifests in three critical ways: non-IID population distributions ($\mathbb{P}(\mathcal{X}^m) \neq \mathbb{P}(\mathcal{X}^l)$), disparate interventions ($\mathcal{T}^m \neq \mathcal{T}^l$), and different accessible variables ($d^m \neq d^l$) across clients, while outcomes ($\mathcal{Y}^m$) may be identical or different. While we discuss FL across institutions in this paper, the same technical approach can be applied within single institutions to unify heterogeneous internal data sources where similar challenges of non-identical covariates and treatments exist.

## 3 METHODOLOGY

In this section we present the details of the proposed framework - FedTransTEE. The right side of Figure 2 shows the workflow of our method used at each site, which includes the following components - i) the covariate encoder that takes the individual subject (patient) features to create a subject-level latent representation (embedding), ii) the treatment encoder that encodes the specific intervention into another latent representation, iii) the cross attention module that models the interaction between the covariate embedding and the treatment embedding, and iv) a predictor that is used to predict the outcome of the treatment. This architecture is carefully designed to handle heterogeneity in covariate and treatment spaces, the need for robust causal relationship modeling and personalization to adapt local variations while leveraging shared knowledge.

### 3.1 COVARIATE ENCODER

At different healthcare sites, patient data is recorded using varied protocols, leading to differences in the types of features collected and how these features are represented. To address this variability, we've designed a covariate encoder that can work with these diverse feature sets for collaboratively learning across sites. Transformers have exhibited remarkable proficiency in learning representations and processing inputs of varying lengths, making them well-suited as the core module for representation learning in the covariate encoder. Each patient covariate vector undergoes processing (discussed in

next section) to generate a sequence of tokens which is fed through a learnable embedder to convert it into a sequence of 256-dimensional embeddings. This sequence of embeddings then proceed through a transformer containing $k$ layers of multi-headed self-attention modules. The entire transformer module enables the acquisition of an effective patient representation, which is then outputted by the covariate encoder. The input-processing unit and the transformer module are collectively known as the covariate encoder and are denoted by $\phi$. Below, we specify the particulars of the input processing and the transformer based representation learning.

**Input Processing** The input processing module takes in the names and values of each covariate for every patient along with the specifications indicating whether the information in each covariate is numerical or categorical and outputs a joint embedded vector (containing the sequence of embeddings) that includes all the patient level features. If $\mathbf{x} = [x_1, x_2, \ldots x_{d_m}]$ is a covariate vector at the $m^{th}$ site, the processed vector, $\mathbf{x}' = f(\mathbf{x}) = [h_1(x_1), h_2(x_2), \ldots h_{d_m}(x_{d_m})]$, is obtained by applying an input processing function $h_i : x_i \to \mathbb{R}^d$ on each of the $x_i$'s. If $x_i$ is a categorical feature, the name of the feature is concatenated with the value of the feature to create a sequence of tokens, which is then tokenized and passed through an embedder to get embeddings. If $x_i$ is a numerical feature, the embedding is obtained for the name of the feature and the embedding is multiplied by the value of $x_i$ to generate the final embedding. The embedder employs a learnable 256-dimensional embedding for each token in the vocabulary and is accessed via a lookup table. Each feature's embedding is then concatenated to obtain $\mathbf{x}'$ and a special learnable [CLS] token is prepended to $\mathbf{x}'$ to obtain the final embedding of $\mathbf{x}$, $\mathbf{e(x)} = [[\text{CLS}], h_1(x_1), h_2(x_2), \ldots h_{d_m}(x_{d_m})]$. This method of processing the input helps in capturing the semantic meaning of the covariates.

**Representation Learning** The initial sequence of embeddings, $\mathbf{e(x)}$, of the covariates are then passed through a transformer network consisting of $k$-layers where each layer is composed of a multi-head self attention module followed by a multi-layer perceptron with residual connection in between and layer normalization at the end. The attention mechanisms encoded in the attention matrix allows the model to focus on different parts of the inputs for learning a better representation and multiple attention heads are used to allow it to attend to different parts of the input sequence simultaneously. We use a 8-head self-attention network in our architecture.

### 3.2 TREATMENT ENCODER

The treatment encoder ($\psi$) is used to convert specific treatments into 256-dimensional embeddings, denoted by $\psi(j)$ for treatment $j$. This component includes an optional treatment information encoder, which processes information like textual descriptions or drug compositions about the treatments to create a representation of the treatment in a high-dimensional space. Following this, there's a learnable two-layer MLP (multi-layer perceptron) with a ReLU activation function in between. If additional details about a treatment are provided, the details are fed into the information encoder and its output is given to the MLP to generate the final 256-dimensional treatment embedding. Alternatively, if no additional information is available, the MLP takes a one-hot representation of the 1-of-$K$ treatments used on specific subjects as input to generate the final embedding. In our experiments (wherever mentioned), we consider the textual descriptions of the treatments from clinicaltrials.gov website as the supplementary information, and utilize pre-trained language models like GPT, BERT, etc. for encoding these descriptions. Other specialized off-the-shelf or pre-trained encoders capable of interpreting specific treatment level information can be plugged in for use in our framework. This treatment information encoder proves particularly valuable in zero-shot scenarios, as it predicts the outcomes of newly introduced and distinct treatments by leveraging existing similarities with treatments already seen by the model, as demonstrated in our experiments later on.

### 3.3 CROSS ATTENTION MODULE AND PREDICTOR

The cross-attention module, $g()$, takes in the treatment and the patient embeddings and learns the interaction between them through a cross-attention transformer block. As opposed to self-attention, the cross attention uses the treatment embedding as the query and the sequence of feature embeddings in the patient embedding as the keys and values. Multiple cross-attentions are employed to facilitate information exchange between parallel treatment and patient representations. We use a transformer block of single layer with 8 attention heads for the cross-attention encoder. Finally, a predictor constructed from a two-layer MLP with ReLU activation non-linearity is used to predict the observed (factual) outcome for the given treatment and the covariate vector, $\hat{y} = f_m(g(\psi(j), \phi(x)))$ for the $m^{th}$ site. Employing a separate predictor allows for personalized modeling within the FL framework. This approach enables collaborative learning of joint treatment and patient embeddings across all

sites, while keeping the prediction head independent to handle diverse outcomes measured across these sites.

## 3.4 LOCAL OPTIMIZATION AND COLLABORATION

The overall training process consists of $E$ communication rounds between the clients and $e$ local training epochs on individual clients. In each communication round, the globally aggregated covariate encoder ($\bar{\phi}$), treatment encoder ($\bar{\psi}$), and the cross-attention module ($\bar{g}$) are broadcasted to all the clients. The clients initialize their corresponding local modules with the obtained global modules and train the entire local model for $e$ epochs, at the end of which the locally trained models are sent to the server where they are again aggregated.

The local optimization at each client involves an alternate minimization procedure where the predictor and the representation learning modules are optimized alternatively on the mean squared error loss between the prediction and the true outcome of the treatment on the given patient. Specifically, at each client $m$, the optimization procedure first updates the predictor $f_m$ by

$$f_m^* = \arg\min_f \mathbb{E}_{(x,t,y)\sim \boldsymbol{D}_m}\left[\left(f(\bar{g}(\bar{\psi}(t),\bar{\phi}(x))) - y\right)^2\right],$$

and then uses the optimized predictor to learn the other components

$$\{\psi_m^*, \phi_m^*, g_m^*\} = \arg\min_{\{\psi_m,\phi_m,g_m\}} \mathbb{E}_{(x,t,y)\sim \boldsymbol{D}_m}\left[\left(f_m^*(g_m(\psi_m(t),\phi_m(x))) - y\right)^2\right].$$

The optimized $\{\psi_m^*, \phi_m^*, g_m^*\}$ are uploaded on the server and aggregated parameter-wise where the parameters of the clients are weighted according to the number of data points present on each client. The global aggregated versions of the models are obtained in the following way -

$$\bar{\psi} = \sum_{i=1}^{N} \frac{n_i}{\sum_{i'=1}^{N} n_{i'}}\psi_i; \quad \bar{\phi} = \sum_{i=1}^{N} \frac{n_i}{\sum_{i'=1}^{N} n_{i'}}\phi_i; \quad \bar{g} = \sum_{i=1}^{N} \frac{n_i}{\sum_{i'=1}^{N} n_{i'}}g_i.$$

The aggregated parameters are sent to the clients again for the next round of training, and the entire procedure continues for a total $E$ number of communication rounds.

## 4 EXPERIMENTAL EVALUATION

In this section, we demonstrate the experimental evaluation of our method and compare it with various baseline approaches. Additionally, we also aim to explore additional questions related to the zero-shot learning capability of the framework, and the interpretability of the approach.

### 4.1 EXPERIMENTAL SETTING

**Datasets and Heterogenity** Our experimental design systematically evaluates model performance across varied datasets with different sizes and varying degrees of heterogeneity - i) Minimal Heterogeneity: We utilize three prevalent semi-synthetic datasets: the Infant Health and Development Program (IHDP) Shalit et al. (2017), 2016 Atlantic Causal Inference Conference (ACIC-2016) Competition Dorie et al. (2017), and Twins Almond et al. (2005) dataset. The experimentation protocol based on semi-synthetic datasets serves two critical purposes: a) the presence of ground-truth data for both factual and counterfactual outcomes enables precise calculation of treatment effects and evaluation across metrics such as PEHE and ATE, and b) their binary treatment and identical covariate setting renders them suitable for fair baseline comparisons; ii) Moderate Heterogeneity: We evaluate on real-world data collected from three randomized clinical trials for intracerebral hemorrhage (ICH) therapy development: ATACHII (NCT01176565), MISTIEIII (NCT01827046), and ERICH (NCT01202864) Ling et al. (2024). This dataset exhibits inherent non-identical covariate spaces across the three locations where these data was obtained, with quantifiable heterogeneity: total unique covariates $|A \cup M \cup E| = 45$, common covariates $|A \cap M \cap E| = 13$, with various partial overlaps ($|(A \cap E) \setminus M| = 18, |(M \cap E) \setminus A| = 2, |M \setminus (A \cap E)| = 2, |A \setminus (M \cap E)| = 7, |E \setminus (A \cap M)| = 3$); iii) High Heterogeneity: We use the CPAD dataset containing Alzheimer's disease or mild cognitive impairment data from 38 Phase II/III randomized clinical trials across 19 distinct study locations with 7 different treatments. This represents extreme heterogeneity: 144 total unique covariates across sites with zero common covariates shared across all 19 sites. More detailed dataset statistics are included in Table 5 in Appendix C.

**Multi-site Simulation** To simulate decentralized federated learning, we partition datasets into multiple clients. The semi-synthetic datasets are partitioned to maintain identical covariate spaces and treatments across clients (for comparisons with baselines) while introducing population heterogeneity through varying treatment/control ratios. The IHDP dataset is divided into 3 sites (240 samples/site), ACIC-16 into 5 sites (960 samples/site), and Twins into 10 sites (1140 samples/site). The real-world ICH dataset naturally divides into three parts corresponding to different hospital locations, each implementing a unique treatment reflecting how this data was actually collected. Similarly, the CPAD dataset encompasses 19 sites reflecting data from distinct study locations. Each client's data is split into training, validation, and test sets (70 : 15 : 15). This design also allows us to evaluate robustness across varying dataset sizes.

**Baselines** We compare FedTransTEE against different baseline methods in three categories - i) Federated ITE estimation methods: iFedTree Tan et al. (2022) and FedCI Xiong et al. (2023) (note that iFedTree only predicts factual outcomes), ii) State-of-the-art ITE estimation methods using CATENets package (Curth). Since these methods don't inherently support multiple sources, we train them in two configurations - a) Local training: Individual models trained separately on each site without collaboration, b) Centralized training: A hypothetical setting where all data is collected at a single site (included for comparative purposes only, not possible in real-world scenarios), to compare against our method.

**Metrics** Given the unavailability of ground-truth treatment effects in real-world data, directly evaluating ITE estimation methods is challenging. Consequently, we employ two distinct sets of metrics for experiments conducted on semi-synthetic datasets and real-world datasets. With the semi-synthetic datasets where both the factual and the counterfactual outcomes are available, we measure and report the Root Mean Squared Error on the factual outcome (RMSE-F), the Error in the Average Treatment Effect ($ATE_\epsilon$), and the Precision in the Estimation of Heterogeneous Effects (PEHE). And for the real-world datasets when only the factual outcome is available we report the RMSE-F, and the difference in Average Treatment on Treated ($ATT_\epsilon$). These metrics are defined in Appendix D.

**Training parameters** All experiments were conducted over 5 repetitions with evaluation on held-out test data. Our architecture employs a covariate encoder with 2 transformer layers (8 attention heads each) and a cross-attention module with 1 transformer layer (8 attention heads), all with hidden dimension 256. For treatment encoding, we utilize BERT encoder Devlin et al. (2019) to generate 786-dimensional embeddings of treatment descriptions when available; otherwise without the information, treatments are represented using one-hot encoding. The federated training process runs for 200 communication rounds with 5 local training epochs per site and early stopping patience of 20 epochs. We use Adam optimizer with learning rate 5e-3 and batch size 128. All hyperparameters for both our method and baselines were tuned on validation sets, and models were trained on a 4-GPU machine with GeForce RTX 3090 GPUs (24GB memory per GPU).

## 4.2 RESULTS

For semi-synthetic datasets (minimal heterogeneity), we report results in Table 1 which shows FedTransTEE significantly outperforms other federated baselines on all semi-synthetic datasets. Secondly, the results in the centralized setting benchmark our method against centralized baselines and serve as an upper bound on achievable performance if all data were available in a single location (impractical scenario), where we observe that the proposed method outperforms all the other methods suggesting its efficiency in predicting ITE on the collective data source as well. Comparing federated versus local training results (Table 1 and 7) demonstrates consistent performance gains through federation, confirming effective knowledge transfer between sites while preserving privacy. For real-world datasets (moderate to high heterogeneity), Table 2 demonstrates FedTransTEE's superior performance on datasets with heterogeneous covariate and treatment spaces. In this scenario, the single-source baselines have to operate in the local learning setting and other FL baselines are inapplicable due to their inability to accommodate multiple treatments across sites. Notably, our method not only exhibits superior average performance but also demonstrates lower variance. This reduced variance is particularly valuable in clinical settings, indicating more reliable performance across diverse institutional contexts regardless of local data limitations.

FedTransTEE maintains consistent effectiveness across varying client dataset sizes (IHDP: 240, ACIC-16: 960, Twins: 1140) and heterogeneity levels. While transformer architectures traditionally require large datasets, our federated approach effectively mitigates data scarcity challenges through collaborative knowledge sharing. The model's strong performance across all experimental conditions,

from minimal to high heterogeneity and from small to large datasets, confirms its versatility for real-world healthcare applications.

Table 1: The table shows performance comparison between our method and the related ITE prediction methods on the held-out test dataset for the semi-synthetic datasets averaged over 5 runs.

| Method | IHDP | | | ACIC-16 | | | Twins | | |
|---|---|---|---|---|---|---|---|---|---|
| | (PEHE) | (ATE$_\epsilon$) | (RMSE-F) | (PEHE) | (ATE$_\epsilon$) | (RMSE-F) | (PEHE) | (ATE$_\epsilon$) | (RMSE-F) |
| FedCI | $1.33 \pm 0.20$ | $0.65 \pm 0.14$ | $2.59 \pm 0.09$ | $2.3 \pm 0.04$ | $1.14 \pm 0.19$ | $1.64 \pm 0.02$ | $0.34 \pm 0.01$ | $0.052 \pm 0.01$ | $0.56 \pm 0.1$ |
| iFedTree | - | - | $2.0 \pm 0.15$ | - | - | $2.53 \pm 0.1$ | - | - | $0.09 \pm 0.001$ |
| FedTransTEE (Ours) | $\mathbf{1.02 \pm 0.01}$ | $\mathbf{0.26 \pm 0.06}$ | $\mathbf{1.77 \pm 0.5}$ | $\mathbf{0.78 \pm 0.1}$ | $\mathbf{0.326 \pm 0.02}$ | $\mathbf{0.728 \pm 0.01}$ | $\mathbf{0.32 \pm 0.01}$ | $\mathbf{0.01 \pm 0.002}$ | $\mathbf{0.09 \pm 0.01}$ |
| (Central) | | | | | | | | | |
| S-Learner$_c$ | $0.93 \pm 0.005$ | $3.7 \pm 0.001$ | $1.22 \pm 0.02$ | $2.6 \pm 0.009$ | $3.4 \pm 0.002$ | $0.61 \pm 0.003$ | $0.32 \pm 0.01$ | $0.01 \pm 0.001$ | $0.08 \pm 0.02$ |
| T-Learner$_c$ | $1.22 \pm 0.02$ | $3.6 \pm 0.01$ | $1.23 \pm 0.04$ | $3.8 \pm 0.01$ | $3.46 \pm 0.001$ | $0.56 \pm 0.005$ | $0.33 \pm 0.01$ | $0.02 \pm 0.001$ | $0.09 \pm 0.003$ |
| TARNet$_c$ | $1.19 \pm 0.002$ | $3.9 \pm 0.01$ | $1.26 \pm 0.02$ | $2.94 \pm 0.01$ | $3.54 \pm 0.001$ | $0.41 \pm 0.001$ | $0.32 \pm 0.001$ | $0.02 \pm 0.001$ | $0.09 \pm 0.005$ |
| FlexTENet$_c$ | $1.19 \pm 0.005$ | $3.9 \pm 0.05$ | $1.2 \pm 0.01$ | $2.83 \pm 0.01$ | $3.5 \pm 0.001$ | $\mathbf{0.34 \pm 0.001}$ | $0.32 \pm 0.001$ | $0.015 \pm 0.001$ | $0.085 \pm 0.001$ |
| HyperITE$_c$ (S-Learner) | $0.91 \pm 0.001$ | $3.9 \pm 0.01$ | $1.19 \pm 0.02$ | $2.6 \pm 0.01$ | $3.5 \pm 0.01$ | $0.44 \pm 0.01$ | $0.32 \pm 0.002$ | $0.02 \pm 0.001$ | $0.085 \pm 0.01$ |
| HyperITE$_c$ (TARNet) | $1.08 \pm 0.04$ | $3.8 \pm 0.003$ | $1.26 \pm 0.009$ | $2.9 \pm 0.02$ | $3.5 \pm 0.02$ | $0.6 \pm 0.01$ | $0.32 \pm 0.01$ | $0.02 \pm 0.001$ | $0.089 \pm 0.02$ |
| FedTransTEE$_c$ (Ours) | $\mathbf{0.90 \pm 0.01}$ | $\mathbf{0.14 \pm 0.1}$ | $\mathbf{0.97 \pm 0.01}$ | $\mathbf{0.78 \pm 0.01}$ | $\mathbf{0.37 \pm 0.10}$ | $0.45 \pm 0.04$ | $\mathbf{0.30 \pm 0.01}$ | $\mathbf{0.01 \pm 0.004}$ | $\mathbf{0.08 \pm 0.002}$ |

Table 2: The table presents a performance comparison between our method and related ITE prediction methods for real-world datasets. The reported metrics in these results are averaged across all clients for each run.

| Method | ICH | | CPAD | |
|---|---|---|---|---|
| | (RMSE-F) | (ATT$_\epsilon$) | (RMSE-F) | (ATT$_\epsilon$) |
| S-Learner$_l$ | $1.32 \pm 0.35$ | $0.22 \pm 0.01$ | $7.4 \pm 3.8$ | $10.1 \pm 0.01$ |
| T-Learner$_l$ | $1.34 \pm 0.41$ | $0.24 \pm 0.02$ | $6.9 \pm 3.2$ | $9.8 \pm 0.02$ |
| TARNet$_l$ | $1.33 \pm 0.44$ | $0.23 \pm 0.01$ | $7.1 \pm 3.04$ | $9.5 \pm 0.01$ |
| FlexTENet$_l$ | $1.33 \pm 0.38$ | $0.23 \pm 0.03$ | $5.6 \pm 0.45$ | $\mathbf{8.1 \pm 0.89}$ |
| HyperITE$_l$ (S-Learner) | $1.30 \pm 0.32$ | $0.22 \pm 0.01$ | $7.45 \pm 3.58$ | $13.4 \pm 1.6$ |
| HyperITE$_l$ (TARNet) | $1.32 \pm 0.28$ | $0.22 \pm 0.01$ | $7.2 \pm 2.7$ | $11.2 \pm 2.1$ |
| FedTransTEE (Ours) | $\mathbf{1.19 \pm 0.09}$ | $\mathbf{0.10 \pm 0.02}$ | $\mathbf{4.6 \pm 0.06}$ | $10.2 \pm 0.02$ |

### 4.2.1 ZERO-SHOT ITE ESTIMATION

We further evaluate the performance of the proposed method in zero-shot testing scenarios. This entails situations where there is no historical data available for a newly designed treatment, or when a new site that seeks to estimate the effects of its treatments is introduced. In such cases, we explore how the proposed framework can be utilized for estimating the effects of the unseen treatments. To enable zero-shot learning, we obtain additional information like a description (or a set of features) for the unseen treatment, and use this information to generate an embedding of the treatment, and pass it to our proposed framework (which was trained on the data and descriptions of the other treatments). To assess this capability, we set up an experiment where we exclude data related to a specific treatment (ATACH-II or MISTIE-III) from the ICH training dataset. We then train the model solely on data from the other two treatments, using treatment descriptions obtained from clinicaltrials.gov (excerpts of which are shown in Table 6). The model is subsequently tested on the data corresponding to the excluded treatment (ATACH-II or MISTIE-III). The results of this zero-shot performance evaluation are presented in Table 3. For comparison, we also include metrics for the same test dataset when it was part of the training procedure, shown under the supervised performance column in Table 3. These results validate that zero-shot capability is achievable within our framework, demonstrating that our encoders effectively capture meaningful relationships both between patient covariates and treatments, as well as among different treatments themselves. This empirical demonstration underscores our framework's effectiveness and reveals promising avenues for its practical application across diverse healthcare settings.

### 4.2.2 INTERPRETABILITY ANALYSIS

**Important covariates for outcome prediction.** We investigate the self-attention mechanism of the covariate encoder to uncover interpretable patterns in the relationships among covariates for predicting outcomes in ICH therapy trials. By examining the attention weights, we identify the features deemed most important by the model for its predictions. The attention weights are calculated

by averaging the attention activations over all patients. In the ICH dataset, four self-attention heads displayed distinguishable patterns of attention weights, as detailed below. **Attention Head 1**: The final embedding (`[CLS]` token) heavily attends to a patient's prior history, particularly racial group (being White or not) and type 2 diabetes. This finding aligns with established clinical knowledge that race and diabetes comorbidity significantly impact outcomes such as functional independence, measured by the mRS score, after ICH onset (Zheng et al., 2018; Woo et al., 2022). **Attention Head 2**: A strong connection is observed between prior vascular conditions and hematoma pathology. Key variables such as intraventricular hemorrhage volume, anticoagulant use, Asian race, prior history of heart failure, and hematoma location are strongly attended to by the final embedding. This suggests that Attention Head 2 captures the pathology of hematoma development and progression, which is a direct therapy target in ICH treatment. **Attention Head 3**: A clear relationship emerges between the final embedding and the Glasgow Coma Scale (GCS), which measures the level of consciousness after a brain injury. This connection is consistent with clinical knowledge, as the GCS score is a critical indicator of brain injury severity and functional independence after ICH onset (Rost et al., 2008). **Attention Head 4**: This head predominantly attends to the NIH Stroke Scale (NIHSS), which indicates stroke severity. The strong focus on the NIHSS score underscores its significance in predicting functional independence after ICH onset. Both GCS and NIHSS are well-known predictors of clinical outcomes following ICH (Hemphill III et al., 2001). Attention matrices corresponding to the two most prominent self-attention heads are visualized in Figure 4 and Figure 5.

Table 3: Test RMSE-F on ICH dataset under the **zero-shot** training protocol.

| Treatment Name | Supervised Performance | Zero-shot Performance | $\Delta$ |
|---|---|---|---|
| ATACH-II | 1.21±0.01 | 1.30±0.06 | ∼0.1 |
| MISTIE-III | 0.82±0.05 | 1.02±0.04 | ∼0.2 |

**Important covariates for each treatment**. We also investigated the cross-attention between the covariate and treatment encoders to identify which covariates are important for outcome prediction under each treatment. Specifically, we analyzed the top covariates with high attention weights for the `[CLS]` token for each treatment. For **ATACH-II**, the key covariates identified, in decreasing order of importance, were initial systolic and diastolic blood pressure (SBP and DBP), GCS score, Hispanic ethnicity, and White race. High SBP is a critical factor in intracerebral hemorrhage (ICH) outcomes, and the ATACH-II trial specifically targeted SBP management to prevent hematoma expansion and improve clinical outcomes. While DBP is less commonly emphasized compared to SBP, it still contributes to cardiovascular stress and influences outcomes. The GCS score is a well-established predictor of ICH outcomes. For **MISTIE-III**, the important covariates identified were platelet count, sodium, potassium, $CO_2$, BUN, APTT, WBC count, and chloride level. These variables are essential for patient recovery following surgical interventions like those in MISTIE-III. For instance, platelet count is critical for assessing bleeding risk and clotting ability; low platelet counts increase the risk of hemorrhage during and after surgery (Ziai et al., 2003). These analyses confirm that the patient representations learned by the covariate encoder and the treatment effect captured by the cross-attention mechanism indeed reflect significant treatment- and disease-level information. A visualization of the cross-attention activations is provided in Figure 3.

## 5 RELATED WORK

This section provides a brief overview of the most relevant prior work in the fields of ITE estimation, and federated learning for healthcare and ITE estimation. We highlight the novel aspects of our work in relation to relevant previous research in Table 4 and include relevant work in federated learning domain in Appendix A due to space constraints.

**ITE Estimation** Due to their ability to handle high-dimensional data and intricate feature interactions, machine learning methods have been increasingly used for ITE estimation. However, most existing methods estimate effects using locally accessible data sources. Broadly, ITE estimation methods can be categorized into two groups - *indirect learners*, which estimate the potential outcomes for each intervention and compute ITE as the difference between the predicted outcomes with and without the intervention, and *direct learners*, which directly model ITE without explicitly estimating the potential outcomes. Among indirect learners, many approaches augment covariates by incorporating intervention information to form the input set $(\mathcal{X}, \mathcal{T})$ for predicting $\mu_1(x)$ and $\mu_0(x)$. These include methods using regression trees (Athey & Imbens, 2016), random forests (Künzel et al., 2017), non-parametric approaches (Curth & van der Schaar, 2021a), and Bayesian methods (Hill, 2011). However,

conventional supervised learning methods that use intervention as a feature often struggle with selection bias inherent in intervention assignment, limiting their direct applicability to ITE estimation. To address this, other approaches build separate models for each potential outcome under the Neyman-Rubin framework. Early works in this category utilized regression-based modeling (Cai et al., 2011), trees, random forests (Foster et al., 2011; Athey & Imbens, 2015), and other techniques (Nie & Wager, 2017; Kennedy, 2020). Recently, the focus has shifted toward neural networks, with the most common strategy involving representation learning from input data followed by intervention-specific predictive models. Notable examples include methods leveraging deep neural networks (Johansson et al., 2016; Alaa & van der Schaar, 2017; Shalit et al., 2017; Atan et al., 2018; Qidong et al., 2020; Zhang et al., 2020; Curth & van der Schaar, 2021b), transfer learning (Bica & van der Schaar, 2022b), and Bayesian approaches (Yao et al., 2018; Hassanpour & Greiner, 2020; Curth et al., 2021; Chauhan et al., 2023). On the other hand, direct learners typically estimate nuisance parameters, which are then used to directly infer treatment effects. Each direct learner defines its own set of nuisance parameters and employs unique methods for their estimation (Wager & Athey, 2015; Powers et al., 2017; Yoon et al., 2018; Kristiadi, 2019). Recently, pair-based approaches, such as learning predictors from pairs of examples with observed outcomes, have also been introduced (Nagalapatti et al., 2024).

**Federated Learning for Healthcare and ITE Estimation** FL has demonstrated significant utility in healthcare (Rieke et al., 2020; Sheller et al., 2020; Prayitno et al., 2021), enabling numerous applications such as predicting adverse drug reactions (Choudhury et al., 2020), stroke prevention (Ju et al., 2020), and mortality prediction (Vaid et al., 2021). Despite these advancements, the federated estimation of causal effects has received limited attention. Xiong et al. (2023) proposed a method to predict average treatment effects by locally computing summary statistics and then aggregating these across sites. Vo et al. (2022) introduced a Bayesian mechanism that integrates local causal effects from different sites to estimate posterior distributions but assumed identical data distributions across sites, which was later extended to handle dissimilar distributions by Vo et al. (2024). Privacy-preserving learning methods (Han et al., 2023a;b) have been proposed for federated settings but typically assume identical covariates across sites, a condition often unmet in practice. Other approaches, such as Yang & Ding (2018); Zeng et al. (2023); Han et al. (2023c), address non-identical covariates by leveraging transportability to transfer causal effects from a source population distributed across multiple sites to a target population. Khellaf et al. (2024) compared meta-analysis with one-shot and multi-shot FL approaches for decentralized data but assumed identical covariates across sites. While some methods accommodate heterogeneous covariates, they often fail to address the challenge of disparate interventions or treatments administered across sites, highlighting a significant gap in the current literature. Recent advancements have investigated the use of **transformers for tabular data** (Wang & Sun, 2022; Zhu et al., 2023; Wang et al., 2024; Hollmann et al., 2025), highlighting their effectiveness for structured data tasks. However, their application to causal inference, particularly in the context of ITE estimation, remains unexplored. This work leverages transformers to address the unique challenges of causal inference, making novel contribution in this domain. Table 4 compares our approach with the most closely related works.

## 6 CONCLUSION

We introduced FedTransTEE, a novel framework for ITE estimation that effectively bridges the gap between healthcare data silos while accommodating heterogeneity in covariates, treatments, and outcome spaces. Our extensive experimentation across semi-synthetic benchmarks and real-world clinical trials demonstrates the framework's superior performance in both federated and centralized settings. The framework's interpretability mechanisms provide clinically meaningful insights validated by domain experts, enhancing trust and adoption potential. The zero-shot capability demonstration enables reliable estimation of treatment effects for previously unseen interventions by leveraging additional treatment information, a comprehensive evaluation of zero-shot robustness across diverse settings remains part of our future work agenda. Our results confirm two fundamental insights - i) collaborative learning across diverse institutions significantly improves treatment effect estimator quality, and ii) strategic use of treatment information enables accurate prediction of novel treatment effects. While the federated mechanism does not entail explicit data sharing, the method may encounter limitations in settings where sharing model parameters outside the client site, even with a trusted server, poses privacy risks. Therefore, a privacy-preserving version of the method will be considered for future work. Also, while this work addresses a critical gap in the literature by providing a novel and empirically validated framework under realistic settings, developing theoretical guarantees on the method will be completed as part of the future work.

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

# Supplement for "Beyond Data Silos : Leveraging Disparate Data Sources for ITE Estimation"

Table 4: Contrasting our method, FedTransTEE, against previous works.

| Method | Addressed Challenges | | | |
|---|---|---|---|---|
| | ITE Estimation | Federated Learning | Heterogeneous Data Sources | Explainability |
| HyperITE | ✓ | ✗ | ✗ | ✗ |
| FlexTENet | ✓ | ✗ | ✗ | ✗ |
| TARNet | ✓ | ✗ | ✗ | ✗ |
| PairNet | ✓ | ✗ | ✗ | ✗ |
| Unipredict | ✗ | ✗ | ✓ | ✗ |
| Xtab | ✗ | ✗ | ✓ | ✗ |
| iFedTree | ✓ | ✓ | ✗ | ✗ |
| FedCI | ✓ | ✓ | ✗ | ✗ |
| **FedTransTEE** | ✓ | ✓ | ✓ | ✓ |

In this supplementary material, we begin by offering further details on the background, followed by an overview of the datasets employed in the experiments. We then provide precise definitions of the evaluation metrics used. Subsequently, additional experimental results are presented, focusing on the local training of the models, along with visualizations related to the interpretability analysis discussed in subsection 4.2.2.

## A  ADDITIONAL RELATED WORK

**Federated Learning** Initially presented as the FedAvg algorithm in the pioneering study by (McMahan et al., 2017), FL has since undergone numerous modifications tailored to address distinct challenges. These adaptations include both global solutions as well as personalized solutions that cater to the non-IID data across clients in a better way. Some key works that obtain global solutions for FL include (Karimireddy et al., 2020; Acar et al., 2021; Chen & Chao, 2021; Collins et al., 2021; Zhang et al., 2021) and that for personalised FL include (Fallah et al., 2020; Li et al., 2020; Makhija et al., 2022; Shamsian et al., 2021). However, it is only very recently that the problem of heterogeneous feature spaces has been explored in the FL setting. Suzuki & Banaei-Kashani (2023) use clustering to figure out similar clients and exchange knowledge within the cluster whereas Rakotomamonjy et al. (2023) use learnt prototypes to align the feature spaces across the clients.

## B  BACKGROUND AND ASSUMPTIONS

Our framework is based on several general assumptions commonly adopted in causal inference. The first assumption, known as the *Stable Unit Treatment Value Assumption (SUTVA)*, requires that each individual's potential outcomes are independent of the potential outcomes of others, and that there is no interference across individuals. The second assumption is *unconfoundedness*, which suggests that there are no unmeasured confounders affecting both the treatment assignment and the outcome. Specifically, the potential outcomes, $\mu_j(x)$, are independent of the treatment variable $\mathcal{T}$, given the observed variables $\mathcal{X}$. The third assumption is *stochasticity* in treatment assignment. Under these assumptions, the ITE of an intervention $j$ for an individual with covariates $\mathcal{X}$ is approximated by the Conditional Average Treatment Effect (CATE), denoted as $\tau_j(x) = \mu_j(x) - \mu_0(x)$. A causal learning model estimates the CATE by predicting $\hat{\mu}_j(x)$ and $\hat{\mu}_0(x)$, which are then used to compute the estimated CATE as $\hat{\tau}_j(x) = \hat{\mu}_j(x) - \hat{\mu}_0(x)$.

We further assume a hierarchical data structure where each hospital/clinical site is drawn from a distribution $P(H)$, within each site $h$, patients are drawn from a site-specific distribution $P(\mathcal{X}|H = h)$, and treatment assignments follow site-specific policies $P(\mathcal{T}|\mathcal{X}, H = h)$. This hierarchical structure allows us to operate under conditional positivity within sites rather than globally: the positivity assumption is required per site, and we do not require all treatments to be available at all sites. This addresses real-world scenarios such as certain drug trials not being available at particular hospitals. Under the assumption that sites are exchangeable, the optimal approach in a frequentist framework for collaboration of models is to perform size-weighted averaging of treatment effects as in FedAvg. By accounting for the hierarchical structure and site-specific variations, the model can provide more accurate personalized treatment recommendations across diverse clinical environments.

## C  DATASETS

We first discuss the additional details of the semi-synthetic datasets. The IHDP dataset involves a binary treatment scenario with real covariates and simulated outcomes. In this dataset, covariates are obtained from both the mother and the child, with explicit child care or specialist home visits treated as interventions. The outcomes are the future cognitive test scores of the children. This dataset includes 25 covariates and approximately ~743 datapoints. The ACIC-2016 dataset, originally introduced for a competition, was sourced from the Collaborative Perinatal Project. It includes 55 covariates and ~4,802 datapoints. The Twins dataset contains 39 covariates and ~11,400 datapoints. It represents a real-world collection of twin births in the US between 1989

and 1991 and includes covariates related to the parents, pregnancy, and birth. In this dataset, birth weight is considered the treatment variable, while one-year mortality serves as the outcome, making it a binary treatment problem with binary outcomes. All three datasets were pre-processed using the processing method employed in CATENets Curth.

For the first real-world dataset, we collect data from three randomized clinical trials for intracerebral hemorrhage (ICH) therapy development: ATACH2 (NCT01176565), MISTIE3 (NCT01827046), and ERICH (NCT01202864). Each hospital location provides patient-level pre-treatment measurements considered as covariates, such as demographics and clinical presentation of the ICH. A treatment variable indicates whether the patient is receiving active treatment or standard of care, and the outcomes are measured using the modified Rankin Score (mRS), which represents the patient's severity. Each of the three treatments is administered at a different hospital location, and all three trials include standard-of-care therapy. The second real-world dataset is obtained from the clinical trials for the Alzheimer's disease. The covariates include the pre-treatment measurements of the patients, and the outcome is the total ADAS-cognitive score of the patient. These real-world datasets are summarized in Table 5.

Table 5: Dataset statistics and brief description for the real-world datasets used for experimental evaluation.

| | # subjects | # covariates | # treatments | # sites | Description |
|---|---|---|---|---|---|
| ICH data | 3279 | 45 | 3 | 3 | Clinical trial data from 3 trials for therapy development of intracerebral hemorrhage, prepared in Ling et al. (2024). |
| CPAD data | 9406 | 144 | 7 | 19 | Clinical trial data for Alzheimer's disease therapy development from 38 Phase II/III trials cpa. |

## C.1 ZERO-SHOT INFERENCE

As discussed in the Experiments section under subsection 4.2.1, we use additional information to test the approach in the zero-shot setting. Since treatment names are available only for the ICH dataset, we can obtain the necessary additional information and test our method on this dataset. The information that we consider involves textual descriptions of the treatments in natural language, and is sourced from treatment descriptions provided on the website clinicaltrials.gov under ARM details for the treatment. On average, these descriptions contain 55 words. To give a clearer understanding of what these descriptions entail, we highlight excerpts from them and present them in Table 6 below.

Table 6: Excerpts of the descriptions obtained from clinicaltrials.gov for ATACH2, MISTIE and Placebo treatments used in zero-shot inference experiments.

| Treatment Name | Description Snippets |
|---|---|
| ATACH-II | Antihypertensive Treatment of Acute Cerebral Hemorrhage II uses early intensive blood pressure lowering. Intravenous nicardipine hydrochloride will be used as necessary (pro re nata or "PRN") as the primary agent in lowering SBP. The goal for the intensive BP reduction group will be to reduce and maintain SBP < 140 mmHg for 24 hours from randomization. |
| MISTIE-III | Subjects randomized to the Minimally Invasive Surgery (MIS) plus rt-PA management arm will undergo minimally invasive surgery followed by up to 9 doses of 1.0 mg of rt-PA (Activase/Alteplase/CathFlo) for intracerebral hemorrhage clot resolution. |
| Placebo | Subjects randomized to medical management will receive the standard medical therapies for the treatment of intracerebral hemorrhage, which includes ICU care only and no planned surgical intervention. |

## D METRICS

In this section, we precisely define the metrics used for evaluating our methods against the baselines. Since, we consider a multi-treatment setting, we show the generalization of the metric definitions for any treatment denoted by $j$ and calculated over $n$ subjects (datapoints). Firstly, we show the metrics applicable when true treatment effects can be calculated by using the factual and counterfactual outcomes available. These metrics are used for evaluation on the semi-synthetic datasets.

$$\text{ATE}_\epsilon(j) = \frac{1}{n} \sum_{i=1}^{n} (\mu_j(x_i) - \mu_0(x_i)) - \frac{1}{n} \sum_{i=1}^{n} (\hat{\mu}_j(x_i) - \hat{\mu}_0(x_i)).$$

$$\text{PEHE}(j) = \frac{1}{n} \sum_{i=1}^{n} \left( (\mu_j(x_i) - \mu_0(x_i)) - (\hat{\mu}_j(x_i) - \hat{\mu}_0(x_i)) \right)^2.$$

Then, we introduce the metrics applicable in real-world settings, where only factual outcomes are available and the true treatment effect cannot be computed.

$$\text{RMSE-F} = \sqrt{\frac{1}{n} \sum_{i=1}^{n} \sum_{j=1}^{K} \mathbb{1}(\mathcal{T}_i = j)(\mathcal{Y}_i(j) - \hat{\mu}_j(x_i))^2}.$$

$$\text{ATT}(j) = \frac{1}{|T_j|} \sum_{i=1}^{n} \mathbb{1}(\mathcal{T}_i = j)\mathcal{Y}_i(j) - \frac{1}{|T_0|} \sum_{i=1}^{n} \mathbb{1}(\mathcal{T}_i = 0)\mathcal{Y}_i(0)$$

$$\text{ATT}_\epsilon(j) = |\text{ATT}(j) - \frac{1}{|T_j|} \sum_{i=1}^{n} \mathbb{1}(\mathcal{T}_i = j)(\hat{\mu}_j(x_i) - \hat{\mu}_0(x_i))|$$

where $|T_j|$ and $|T_0|$ represents the count of patients receiving treatment $j$ and treatment 0 respectively.

# E  ADDITIONAL EXPERIMENTS

This section provides additional experimental results to complement those presented in Table 1. These results are obtained in the training protocol of local model training. This setting is highlighted because when the methods do not suggest a way of collaboration across multiple sites, each site can only train its own local model for ITE estimation using its respective dataset. For comparison, we also show the result obtained if our method FedTransTEE under federated learning protocol is used instead.

Table 7: The results in the table show performance comparison between our method and the related ITE prediction methods on the held-out test dataset for the local setting on semi-synthetic datasets.

| Method | IHDP | | | ACIC-16 | | | Twins | | |
|---|---|---|---|---|---|---|---|---|---|
| | (PEHE) | (ATE$_\epsilon$) | (RMSE-F) | (PEHE) | (ATE$_\epsilon$) | (RMSE-F) | (PEHE) | (ATE$_\epsilon$) | (RMSE-F) |
| (Local) | | | | | | | | | |
| S-Learner$_l$ | 1.11 ± 0.02 | 3.3 ± 0.06 | 1.83 ± 0.09 | 2.2±0.01 | 2.9±0.02 | 2.3±0.1 | **0.32±0.01** | 0.02±0.001 | 0.10±0.02 |
| T-Learner$_l$ | 1.5 ± 0.06 | 3.58 ± 0.13 | **1.55 ± 0.1** | 2.8±0.02 | 3.3±0.02 | 1.4±0.01 | 0.35±0.001 | 0.04±0.001 | 0.11±0.001 |
| TARNet$_l$ | 1.22 ± 0.02 | 3.7 ± 0.06 | 1.59 ± 0.1 | 2.77±0.01 | 3.3±0.02 | 3.4±0.02 | 0.35±0.001 | 0.03±0.01 | 0.11±0.005 |
| FlexTENet$_l$ | 1.22 ± 0.2 | 3.7 ± 0.06 | 1.6 ± 0.07 | 2.6±0.02 | 3.34±0.01 | 1.7±0.01 | 0.33±0.001 | 0.02±0.001 | **0.09±0.002** |
| HyperITE$_l$ (S-Learner) | **1.02 ± 0.03** | 3.59 ± 0.1 | 1.87 ± 0.23 | 2.2±0.01 | 3.1±0.02 | 2.84±0.05 | **0.32±0.001** | 0.02±0.001 | 0.11±0.004 |
| HyperITE$_l$ (TARNet) | 1.06 ± 0.02 | 3.8 ± 0.06 | 1.6 ± 0.07 | 2.7 ± 0.02 | 3.1 ± 0.03 | 3.4 ± 0.01 | 0.33 ± 0.001 | 0.03 ± 0.002 | 0.10 ± 0.002 |
| FedTransTEE (Ours) | **1.02 ± 0.01** | **0.26 ± 0.06** | 1.77 ± 0.5 | **0.78 ± 0.1** | **0.326 ± 0.02** | **0.728 ± 0.01** | **0.32 ± 0.01** | **0.01 ± 0.002** | **0.09 ± 0.01** |

## E.1  ATTENTION VISUALIZATIONS

We visualize and analyze the cross-attention and self-attention heads' activations obtained during the learning process of FedTransTEE. This allows us to identify the covariates that are crucial in predicting outcomes for different treatments and to explore the relationships between these covariates. Such examination underscores both the explainability and interpretability of our approach. While detailed analysis is provided in subsection 4.2.2, the visualizations are included below.

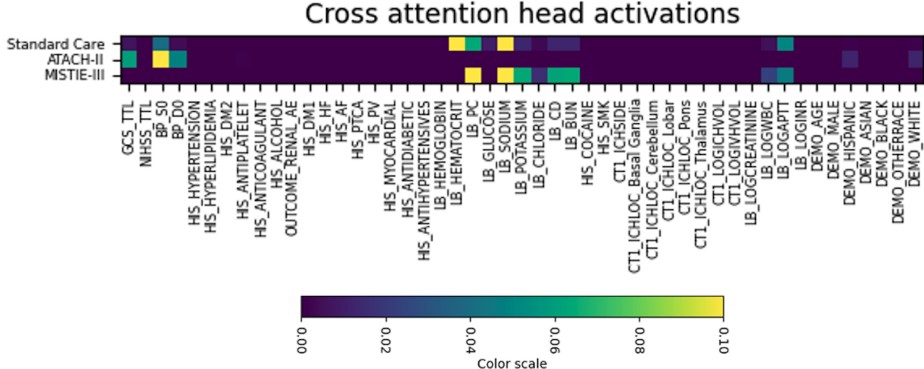

Figure 3: Visualization of the activations of cross-attention head obtained while learning on the ICH dataset.

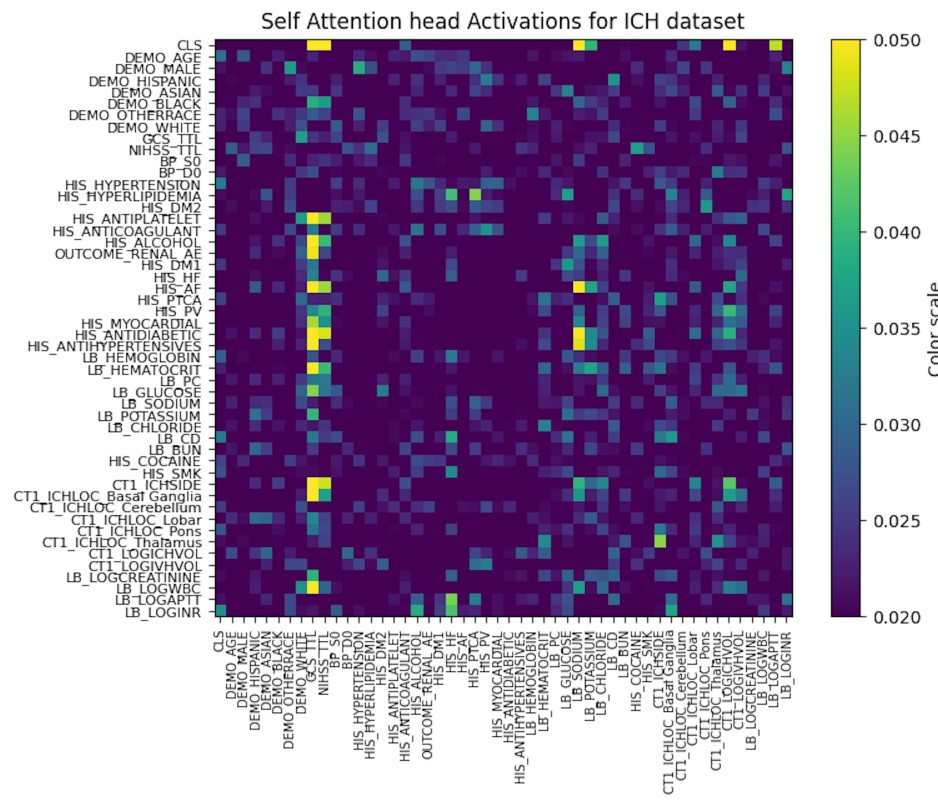

Figure 4: Visualization of the activations of first self-attention head of the covariate encoder obtained while learning on the ICH dataset.

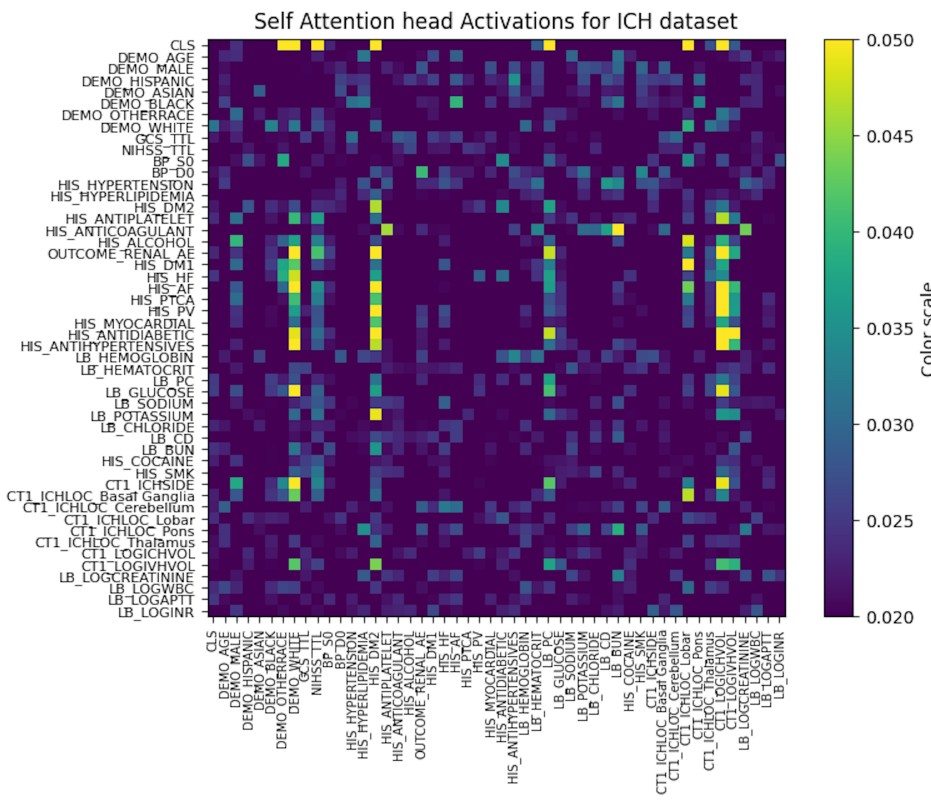

Figure 5: Visualization of the activations of second self-attention head of the covariate encoder obtained while learning on the ICH dataset.

