# OpenReview forum: "Beyond Data Silos : Leveraging Disparate Data Sources for ITE Estimation"
_ICLR.cc/2026/Conference — Submitted to ICLR 2026_

### Official Review · Reviewer_Q5R6 · 2025-10-15

**Soundness:** 2
**Presentation:** 1
**Contribution:** 2
**Rating:** 2
**Confidence:** 4

**Summary:**

In this paper, authors have proposed a novel framework for individualised treatment effects (ITE) estimation under a new and complex setting where there is heterogeneity in covariates, treatments, and outcome spaces. Specifically, they accommodate heterogeneity even in treatment spaces where different hospitals/sites might be using different treatments in addition to having different covariates. This problem is solved in federated learning settings. Under this complex setting, they employ a transformer as the key model for processing heterogeneous features and treatments whose parameters are learnt through sharing parameters across client sites while having site-specific prediction layers. It is interesting to note the way they combined heterogeneous features and treatments from different sites, which also enables the framework to be utilised in a zero-shot learning for evaluating new treatments. Additionally, they evaluated the proposed method across well-established benchmarks as well as real-world RCT trial datasets, all of which show the effectiveness of the method. It is also noted that the attention mechanism highlights some important features for accurate predictions as well as corresponding to different treatments, which have been confirmed by the clinicians for their validity and hence showing the importance of the proposed method.

**Strengths:**

1. The authors have discussed a **novel problem setting**, as evidenced by Table 4, for ITE estimation considering heterogeneity in covariates, treatments and outcome subspaces. Specifically, this consider a setting where different sites not only have different covariates but also may have different treatments given to the subjects, which make it different from the existing work. So, this is ITE estimation in a federated learning setting. I think authors haven't sufficiently stressed this novel aspect of their work. Maybe they should first stress novelty on the problem setting and then on the framework.

2. The other interesting novelty comes from developing a **novel framework**, powered by the transformer architecture, to solve the complex problem settings. I think this is very unique and challenging, the way they handled heterogeneity in covariates and treatments at different federated learning clients. This is because for sharing model parameters, the model should be the same, and it is only this clever approach which made it happen otherwise with heterogeneity, one won't be able to do that (however, alas! this idea is let down due to lack of clarity - discussed below).

3. The authors also claimed that the proposed framework can work in a zero-shot setting. This would enable evaluation of new treatments, which is a very important feature of the model for the field.

4. Since the framework is attention based so it also enables interpretability of results. Authors have claimed that the attention finds clinically relevant features wrt outcomes as well as treatments which were validated by the clinicians.

5. There is sufficient validation across existing benchmarks as well as real world RCT dataset.

**Weaknesses:**

My **major concern is clarity** of this paper, as discussed below. All the key ideas of the paper look magical without explanations.

1. The central point of the paper is that the paper discusses a novel problem setting and then proposes a novel framework to handle this complex and practical problem setting. Moreover, specifically, in a federated learning setting, each client may have different covariates, treatments and outcomes. Now, to share model parameters under a federated learning setting, the model should have the same architecture. However, the authors did not provide sufficient explanations about how input data is processed that despite having different covariates and treatments, the model architecture remains the same with the same input processing. So, the key novelty is let down by a lack of clear descriptions about input processing. This was the key part of the paper, so a visual presentation should have been given for the framework.

2. Authors claim that the proposed model can be used in zero-shot setting for predicting outcomes under new interventions but there isn't sufficient discussion on how this would be enabled; rather, they refer the reader to experiments. They must explain, at least intuition, how this will be enabled. This way, another key idea of the paper is let down by a lack of explanations.

3. The proposed model looks to add a massive computational workload as compared to baselines, which is never discussed/acknowledged in the paper. This is also reflected in its need for GPU-based machines for running experiments.

The way inputs are processed, it looks like it is adding a lot of computations because it processes names of the features in addtion to their values, requiring NLP processing. Secondly, for each feature, multiple tokens are created, each of which has dimensions of 256. So, a proper discussion on the advantages and disadvantages of their input processing should be covered. There should be an analysis of the value added by this approach against the overhead added.

4. FL literature has developed several new methods since the first work. Why the most recent approaches are not discussed and used for weight aggregation? Also, there is no discussion about the baselines for the local setting and how they were selected. Since the paper claims to propose new problem setting so we don't expect much baselines for the federated setting but for local setting they have missed some important baselines, e.g., there are some work based on transformers - why are they not included? Similarly, there was a method from van der Schaar lab (I guess, it is cited in the paper) to deal with heterogeneous covariates -- why is that not compared in the local setting, rather you compared simple homogeneous setting there? I think a massive model will most likely win over simple baselines as used in the paper. Did you discuss their parameters anywhere in the paper?

5. There is no explanation for using an alternate optimisation approach at client sites. Many key ideas are left for the reader to guess.

This way, the key ideas are overshadowed by a lack of clarity. So, I recommend rejection for the current version of the paper.

**Questions:**

N/A

---

### Official Review · Reviewer_Gwic · 2025-10-27

**Soundness:** 1
**Presentation:** 1
**Contribution:** 2
**Rating:** 2
**Confidence:** 4

**Summary:**

The article tackles the very hard problem of vertical and horizontal federated learning for individual treatment effect estimation.

**Strengths:**

Originality of proposing transformers especially with multimodal data for causal inference. The related work section is well done (although appears at the end of the paper?) and the setting is clear.

**Weaknesses:**

Unclear causal assumptions: The paper does not clearly specify whether the unconfoundedness assumption is required within each site individually or across the union of all feature spaces. Moreover, the “stochasticity” assumption mentioned is not defined or justified.

Missing discussion on vertical federated learning: The manuscript does not address key challenges associated with vertical federated learning for causal inference, such as aligning feature spaces or estimating treatment effects when covariates are split across institutions.

Lack of theoretical guarantees: There is no discussion of expected consistency or identification theorems for the estimated ITEs. In particular, it remains unclear how the model ensures identifiability of treatment effects across institutions or whether any overlap in latent representations is assumed.

Presentation issues: Several tables extend beyond the page margins, which impairs readability and should be corrected.
Clear lack of expected consistency theorems for the ITEs. We need identification across institutions. Are assumptions made about overlap in latent representations?

Incomplete baseline coverage: While federated baselines such as FedCI and iFedTree are included, comparisons to more recent multimodal or transformer-based ITE estimation methods are notably missing.

Some federated baselines (e.g., FedCI, iFedTree) are included, but comparisons to recent multimodal or transformer-based ITE models are missing.

I think this paper could be a much more valuable contribution if it was properly positioned as a transformers adaptation to causal inference

**Questions:**

What are the computation costs of your methods?

What safeguards are proposed to prevent potential information leakage through model parameters in federated settings?

Are assumptions made about overlap in latent representations?

Could the framework integrate causal transportability theory or meta-learning to further improve cross-site generalization?

---

### Official Review · Reviewer_uBBN · 2025-10-28

**Soundness:** 2
**Presentation:** 2
**Contribution:** 2
**Rating:** 4
**Confidence:** 2

**Summary:**

The paper proposed _Federated transformers for treatment effect estimation_ (FedTransTEE), a framework that consists of the following components:

1) a covariate encoder, which is a transformer that leverages tokenized labels of the features and the values of the table. (shared)
2) a treatment encoder, which embeds both the description and the value of the treatment into a 256 dimensional vector space. (shared)
3) a cross attention module that combines covariate embedding with treatment embedding. (shared)
4) a predictor that maps the representation that is the output of the cross attention module to the target outcome of the specific site. (not shared)


The proposed framework paper aims to estimate ITEs across multiple sites (hospitals, data centers, trials, etc) even when:
1) covariates are not IID.
2) covariates are not the same: different dimensionality and different semantics.
3) outcome definition differ accross sites.
4) the treatment to evaluate is not the same across sites.
5) data pooling is not feasible due to privacy or other constraits.

The authors claim that this framework will reduce the bias in estimating individual treament effects _even_ when all the previous conditions hold. The framework is evaluated over 1) semisynthetic data used for benchmarking in causal inference, which is divided in several sites, 2) over three random control trials with some shared covariates and 3) over 38 random control trials with no shared covariates.

Results reflect that FedTransTEE achieves consistently better results in the estimation of causal effects (only in semi-synthetic datasets), in predicting the factual outcome and in predicting the average treatment effect. This improvement is shown across all datasets, in both centralized and federated settings.

**Strengths:**

- The paper proposes a FL architecture for ITE estimation that is coherent from a _Representation learning_ point of view. The framework can plausibly reduce variance and improve sample efficiency, and it can leverage label/description information.

- The framework does not require strict feature alignment or identical interventions. The empirical results report performance gains over baselines under clinically realistic heterogeneity.

- The framework supports zero-shot learning, which allows to estimate outcome effects of a new treatment based on its description.

**Weaknesses:**

- Evaluation over different outcomes have not been carried out. One of the claims of the paper is that the framework can model different outcomes in each site, but that is not proven either theoretically or with experimental evaluation.

- The metrics applied to real-world experiment do not provide enough insights about causal inference. First, RMSE-F is a regression metric that do not reflect causal effects. Second, the original data is randomized, therefore potential outcome estimation is not needed to estimate the average treatment effect. Comparing the methods with observational ATT only reflects biases in the factual regression, no counterfactual regression.

- When sites have disjoint covariates and disjoint treatments, there is no clear mechanism that forces alignment of confounders or treatment semantics across sites; at that point the method behaves more like multitask federated representation learning than true cross-site causal transport.

- The zero-shot causal effect estimation is over-stated with the current experiments. First, metrics given are only factual, so causal effects have not been validated. Second, the table is not informative enough: if the treatment is not informative about the outcome, then the absence of the treatment would not affect drastically to the outcome prediction. The table should compare the performance of the model without federated learning, to see if the 'zero-shot' strategy helps.

- While the evaluation in Table 2 include different sites that have different distribution of the treatment (`line 273`), the methods use for comparison, FedCI and iFedTree are not prepared for that setting. Other methods can be used for a fairer comparison: [1, 2]

> Summary

The ideas proposed in this paper are very interesting. Leveraging extra information about labels and sharing encoders across sites may help to the prediction of causal effects. However, the results over real-world data are not conclusive enough for me to recommend the acceptance of the paper. There are some claims that are over-stated and not proven either theoretically or experimentally, as the zero-shot capabilities, the hability of handling with different outcomes in each site, or causal identifiability. In addition, I have accumulated many questions (below) that should be addressed. Although by themselves those questions would not make me to recommend a rejection, the sheer number of them tips the balance towards rejection. I would be happy to raise my score if all questions and more experiments are added, and claims are re-stated in order to adjust the experimental evaluation.

> References

[1] Vo, T. V., Bhattacharyya, A., Lee, Y., & Leong, T. Y. (2022). An adaptive kernel approach to federated learning of heterogeneous causal effects. Advances in Neural Information Processing Systems, 35, 24459-24473.

[2] Almodóvar, A., Parras, J., & Zazo, S. (2024). Propensity Weighted federated learning for treatment effect estimation in distributed imbalanced environments. Computers in Biology and Medicine, 178, 108779.

**Questions:**

I have some questions or some minor concerns that could be addressed. Some of them are oriented to improve readability.

- Could you provide a formal definition of what a _disparate data source_ (or _disparate systems_ in `line 50`) is? It is not clear what do the authors mean by that term, although I can infer its meaning from the rest of the text.

- I have not found any reproducibility statement in the paper. Are the authors planning to sumbit the code to any public repository, or share it somehow? It would help to increase the adoption of its contribution.

- In causal inference, the term _intervention_ usually refers to the _act of intervening_. That is, the action of changing the value of the treatment variable, to evaluate the changes in the outcome. Therefore, I suggest the authors to change the name of 'intervention' by 'treatment' when refering to the random variable (e.g. `line 106`), and keep 'intervention' when refering to _the act of intervening_.

- In ´line 128` the definition of the potential outcome fucntion as the conditional expectation is a decision taken assuming zero-mean additive noise, which could not be the case. I just wanted to be sure that authors understand this particularity.

- _Stochasticity_ assumed in `line 133` is usually refered as _positivity_ and I think that are not exactly the same. Specially, _stochasticity_ is only the same as _positivity_ if the conditional distribution of the treatment has density in all the range. That is, a distribution $P(T|X)$ can be stochastic and, still, has zero density in some points, which may violate positivity.

- The notation in `lines 138-144`should be revised. Authors should differentiate between measurable spaces and random variables. When refering to disparate interventions ($T^l\neq T^m$), does that mean that the support is different? Or does it mean that the random variable is different? If only the support changes, maybe $Supp(T^m) \neq Supp(T^l)$ is clearer.Some light should be given. The same happens with the accessible variables: $d^m$ is only a natural number. Therefore, when we have different $d$, do the models share $\min (d^m, d^l)$? Is not clear _in the notation_ if the random variables $X^l, X^m$ can be different.

- Do the IHDP datasets used in section 4 belong to the setting A or the setting B in (Hill, 2011)?

- Is it necessary to mantain an acronym in the title? some readers may not be familiar with the term.

- Tables are reportred with _averaged_ results. However, I did not find what the intervals mean, and more importantly, what bold numbers mean. Usually, bold means statistical difference. However, no significance test has been reported.

- Table 1 is somehow strange. The centralized version of FedTransTEE achieves much lower errors compared to other baselines, especially in ATE metric. Is there any justification for that? An ATE error of 3.8, 3.9, etc., of baselines in IHDP, knowing that the true ATE is 4, seems to bee too high (Hill, 2011)[1].

- In appendix, `line 745`states that 'sites are exhangeable'. Can you go deeper on that assumption? How can be sites exchangeables if ditributions are heterogeneous?

- When there are no shared treatment and covariates, what reasons do we have to think that the joint training improves the ITE estimation?

- Can authors formalize when sharing the encoders help ITE estimation, rather than risking negative transfer?



> References

[1]Hill, J. L. (2011). Bayesian nonparametric modeling for causal inference. Journal of Computational and Graphical Statistics, 20(1), 217-240.

---

### Official Review · Reviewer_mLKv · 2025-10-29

**Soundness:** 2
**Presentation:** 1
**Contribution:** 2
**Rating:** 2
**Confidence:** 3

**Summary:**

The paper presents FedTransTEE, a transformer-based federated learning framework for Individual Treatment Effect (ITE) estimation across heterogeneous data sources. The model integrates a covariate encoder, treatment encoder, and cross-attention module to enable learning across sites with non-overlapping features and treatments. Experiments on semi-synthetic and real-world clinical datasets show improved performance over baselines, as well as some interpretability and limited zero-shot generalization to unseen treatments.

**Strengths:**

- The idea of combining federated learning and transformer-based representation learning for CATE estimation is novel and practically meaningful.
- The proposed architecture is modular and flexible, capable of handling heterogeneous covariates and treatment spaces.
- The interpretability analysis adds value to clinically meaningful patterns.

**Weaknesses:**

- Despite technical novelty, the paper lacks causal transparency. In ITE estimation, especially with observational or multi-source data, clear and explicit causal assumptions are essential. The paper does not define the target population or clarify whether it aims to estimate site-specific or pooled treatment effects.
- Identification assumptions such as exchangeability across sites are not rigorously stated or discussed. The unconfoundedness assumption is misstated and should be expressed in terms of potential outcomes rather than conditional means.
- In observational studies, the treatment assignment mechanism is unknown, yet the paper does not address the resulting bias or model misspecification risks.
- There is no discussion of conditional outcome drift across data sources, which are key threats to validity in multi-source causal inference.
- There are no formal theoretical guarantees on identifiability, consistency, or bias bounds, leaving unclear under what conditions the proposed estimator produces valid causal effects.
- Overall, modern causal inference practice requires transparency about assumptions, assessment of model misspecification, and explicit discussion of bias sources. The paper’s omission of these elements limits the interpretability and credibility of its causal claims.

**Questions:**

1. What is the exact target estimand, a site-specific, pooled, or transportable CATE?
2. What identification assumptions are needed for valid cross-site estimation in your setting?
3. How is bias handled when (i) treatment assignment mechanisms are not randomized and unknown, (ii) model is misspecified, and (iii) there is conditional outcome drift across data sources?

---

### Meta-Review · Area_Chair_opge · 2026-01-05

**Summary:**

No reviewer will change the score.This paper proposes FedTransTEE, a federated transformer-based framework for ITE estimation across heterogeneous sites with non-overlapping covariates and treatments. Reviewers agree that combining federated learning with transformer-based representations for treatment effect estimation is novel and practically motivated. The architecture is flexible, modular, and empirically shows performance gains in several semi-synthetic and clinical settings. The use of treatment descriptions and the attempt at zero-shot generalization are also interesting directions.

However, the paper suffers from fundamental weaknesses in causal formulation and validation. Most critically, the work lacks causal transparency: the target estimand (site-specific, pooled, or transportable CATE) is not clearly defined, and key identification assumptions are either misstated or not rigorously discussed. There is no treatment of bias arising from unknown treatment assignment mechanisms, model misspecification, or conditional outcome drift, which are central challenges in multi-source observational causal inference. As a result, it is unclear under what conditions the proposed estimator yields valid causal effects.

Empirically, while improvements are reported, the evaluation does not convincingly support the causal claims. Several metrics reflect factual outcome prediction rather than counterfactual accuracy, real-world datasets are randomized where causal estimation is unnecessary for ATE, and important claims (e.g., handling different outcomes across sites and zero-shot causal generalization) are overstated relative to the evidence provided. Comparisons are also sometimes unfair, as some baselines are not designed for the evaluated heterogeneity settings.

Overall, despite promising ideas and a reasonable representation-learning framework, the lack of clear causal estimands, identification guarantees, and convincing causal evaluation limits the credibility of the paper’s claims. A substantial revision clarifying causal assumptions, addressing bias and transport issues, and strengthening experimental validation would be necessary for reconsideration.

**Reviewer Concerns:**

All concerns are not addressed.

**Reviewer Scores:**

No reviewer will change the score.

---

### Decision · Program_Chairs · 2026-01-26

Reject